# Ready for Prime Time? Dendritic Cells in High-Grade Gliomas

**DOI:** 10.3390/cancers15112902

**Published:** 2023-05-25

**Authors:** Claire A. Conarroe, Timothy N. J. Bullock

**Affiliations:** Department of Pathology, School of Medicine, University of Virginia, Charlottesville, VA 22903, USA; tb5v@virginia.edu

**Keywords:** dendritic cells, glioma, glioblastoma, immunotherapy, T cell priming

## Abstract

**Simple Summary:**

Although the immune system can mount effective responses against antigens within the CNS, this does not occur in the context of high-grade gliomas. Considered to be the primary antigen presenters, dendritic cells (DC) are essential for initiating antitumor immune responses through tumor antigen and presentation to T cells and through the secretion of chemokines for T cell migration. Many immunotherapies depend on antigen presentation by dendritic cells, yet dendritic cells are understudied in the specific context of brain tumors such as high-grade gliomas. Accordingly, this review summarizes the presence and function of dendritic cells within the tumor microenvironment of high-grade gliomas with special consideration of therapeutic opportunities.

**Abstract:**

High-grade gliomas are malignant brain tumors, and patient outcomes remain dismal despite the emergence of immunotherapies aimed at promoting tumor elimination by the immune system. A robust antitumor immune response requires the presentation of tumor antigens by dendritic cells (DC) to prime cytolytic T cells. However, there is a paucity of research on dendritic cell activity in the context of high-grade gliomas. As such, this review covers what is known about the role of DC in the CNS, DC infiltration of high-grade gliomas, tumor antigen drainage, the immunogenicity of DC activity, and DC subsets involved in the antitumor immune response. Finally, we consider the implications of suboptimal DC function in the context of immunotherapies and identify opportunities to optimize immunotherapies to treat high-grade gliomas.

## 1. Introduction

High-grade gliomas (HGG) refer to a set of highly aggressive and lethal brain tumors that arise from glial cells. HGG are the most frequent primary CNS malignancy and include astrocytoma, oligodendroglioma, and glioblastoma (GBM), as classified by the updated World Health Organization guidelines [1]. The current standard of care leaves patients with a median survival time of only fifteen months and a five-year survival rate of merely 6.8%, indicative of an urgent clinical need for improved treatment regimens [2,3]. The standard treatment regimen consists of temozolomide chemotherapy, radiotherapy, and tumor resection [2]. Although this combination can acutely limit tumor progression, HGG are diffuse throughout the brain and inevitably recur as a result of malignant cells that remain in small numbers [4,5]. Standard treatment for HGG has not changed in nearly two decades, despite advances in our understanding of cancers, neuroimmunology, and therapeutic options.

Immunotherapies are one of the promising options available, given that they can leverage the patient’s immune system to recognize and eliminate HGG cells. Although the diffuse sections of HGG are challenging to isolate for treatment with radiotherapy or resection, immune cells have the potential to target them more specifically. Effective immunotherapies can also induce long-term immune memory, which may prevent or slow the recurrence that currently plagues HGG patients. Despite success in treating other types of cancer, clinical trials using inhibitor checkpoint blockade (ICB), vaccination strategies, and CAR T cells have yet to show substantial benefit in HGG patients. However, relatively little is known about the immunological landscape of HGG, which constrains our ability to make advances with immunotherapy for these diseases compared to other more commonly and deeply studied extracranial tumors.

Fortunately, the field of neuroimmunology is quickly growing, and advancements in basic research will be critical for the identification and optimization of impactful immunotherapeutic regimens. This is especially important considering patients whose tumors fail to respond to immunotherapies and the potentially toxic side effects that accompany their delivery. Antigen presentating cells (APC), specifically dendritic cells (DC), are of particular interest due to their role in initiating and sustaining the T cell responses that are required for substantive and durable tumor control. Herein, we will review foundational and emerging studies that investigate the potential for DC to promote antitumor immunity in the CNS and the obstacles that prevent this in the context of HGG.

## 2. Overview of DC Function in Antitumor Responses

The innate and adaptive arms of immunity must function cohesively to coordinate a robust immune response. For this reason, DC are critical mediators of immunity. In the fifty years since DC were first described by Ralph Steinman, many DC subsets have been observed to infiltrate a wide variety of tumors, albeit in relatively low proportions [6]. Although the definitions of these DC subsets vary with time and the method used for identification, apart from plasmacytoid DC (pDC) they generally share the ability to induce adaptive immune responses from naïve T cell precursor populations. Conventional DC (cDC) accomplish this by providing essential context beyond TCR engagement, largely through a combination of costimulatory molecule expression and cytokine secretion. For the purpose of this review, we will provide a cursory overview of DC function in cancer immunity below, as there are many excellent in-depth reviews of DC in the context of cancer available [7].

DC are classic professional antigen presenting cells (pAPC) that are either poised in the tissue to phagocytose antigens and migrate to draining lymph nodes or are resident in the lymph nodes where they capture cell-free antigens from lymphatic drainage. DC process acquired antigens for presentation on MHC class I molecules for presentation to CD8^+^ T cells, and MHC class II molecules for presentation to CD4^+^ T cells. In addition to this fundamental role in stimulating MHC-peptide specific T cells, when appropriately activated, DC provide critical costimulatory signals that are necessary to promote T cell cycling and expansion, rather than the induction of anergic unresponsiveness.

A heterogeneous population, DC are further defined into four subsets, each with unique characteristics. cDC1 (defined by XCR1 and DNGR-1 expression) are essential for many antitumor immune responses due to their ability to cross-present tumor antigen to cytotoxic CD8^+^ T cells and to secrete significant amounts of IL-12. Cross-presentation serves as a critical link for priming CD8^+^ T cells, as most cells that express the MHC class I-ligand that is recognized by CD8^+^ T cells do not express the critical costimulatory molecules necessary for robust CD8^+^ T cell activation and expansion. This makes the role of cDC1 especially relevant to antitumor responses since they can uniquely redirect exogenous, phagocytosis-acquired antigens to this pathway for presentation to CD8^+^ T cells [8,9]. cDC1 also contribute to the antitumor response through antigen presentation to CD4^+^ T cells, resulting in the induction of CD40L on T cells, which leads to cDC1 licensing and a subsequently enhanced CD8^+^ T cell response [10]. Moreover, cDC1 are the primary producers of IL-12, a cytokine secreted during antigen presentation that induces Tbet, a transcription factor that empowers effector differentiation in CD8^+^ T cells and polarizes CD4^+^ T cells toward a Th1 phenotype with enhanced IFN-γ production and proliferative capacity [11].

A second subset, cDC2 (commonly, though not perfectly, defined by CD11b and SIRPα expression), also contribute to the antitumor response, primarily through antigen presentation to CD4^+^ T cells via MHCII [12]. Unlike cDC1, the precise role of cDC2 in anti-tumor T cell responses has been more challenging to define, since deletion of single genes involved in cDC2 differentiation does not thoroughly ablate cDC2 in mouse models [13]. Differentiated from infiltrating monocytes under inflammatory conditions, monocyte-derived DC (moDC) appear to overlap with cDC in function yet do not have the same cross-presentation capacity as cDC1 [10]. pDC represent a fourth subset of DC that uniquely differentiate from lymphoid rather than myeloid precursors [14]. Generally dispensable for antigen presentation and T cell priming, pDC are better known for secreting large quantities of Type-1 IFN upon TLR stimulation, which can support cDC-mediated initiation of primary T cell responses. However, tumor infiltrating pDC do not exhibit this function, likely a result of the immunosuppressive factors in the TME [15]. Instead, tumor-infiltrating pDC can recruit Tregs, a highly immunosuppressive subset of CD4^+^ T cells [16]. Overall, DC represent a heterogeneous population of cells with both immunogenic and tolerizing potential within the TME and make logical immunotherapeutic targets due to their ability to promote antitumor immunity, as described in more detail below.

In addition to the subtypes of DC, the character of their interaction with T cells can determine the type of immune response that ensues. DC can be immunogenic or tolerogenic depending on the milieu of their environment and maturation status. Upon priming in the presence of IL-12, which is largely expressed by DC, activated naive T cells differentiate into Th1 cells that produce IFN-γ, a cytokine critical for imposing an inflammatory cytokine signaling loop [17]. In the absence of stimulation via PRRs or cytokines such as Type 1 IFN, DC may still phagocytose and present antigens to T cells; however, these DC do not upregulate expression of MHC or costimulatory molecules. This can lead to T cell anergy and tolerization to the antigen presented [18,19]. Immature DC are also more likely to be tolerogenic, likely due to underdeveloped antigen processing pathways or low level of MHC and costimulatory molecules after antigen uptake; however, mature DC can also be tolerogenic depending on the stimulus [20,21]. Within the context of tumors, it is established that DC tend to have reduced antigen presentation capacity and an immature phenotype [22,23,24]. The HGG TME is characteristically immunosuppressive, with high levels of TGF-B, VEGF, prostaglandins, and IL-10—all factors that can inhibit DC activity as illustrated in Figure 1 [25,26,27,28,29,30,31].

## 3. What Is the Immunological Potential of DC in the CNS?

As an essential yet sensitive organ system, the CNS requires tightly controlled protection against pathogens and disease to avoid pathology. However, the CNS has historically been considered immunologically “privileged” as a result of scarce immune cell presence, perceived lack of lymphatic vessels, and early experiments involving tumor transplantation. In these early experiments, tumors grew when implanted into the brains of immunocompetent rodents but were rejected when implanted in the periphery, inferring location-dependent immunogenicity. Interestingly, when co-transplanted in the brain and a peripheral location, both tumors were rejected, indicating that extracranial immune activity can participate in intracranial tumor rejection. These studies ultimately define the CNS as a tolerogenic organ in which immunity is not generated. However, the discovery of a lymphatic drainage network and effective immune responses to CNS-infiltrating pathogens has led to reconsideration of the CNS as immunologically unique [32,33]. Although DC do not ordinarily reside in healthy brain parenchyma, they mediate immune surveillance in conjunction with border-associated macrophages (BAM) along the brain border in the leptomeninges, dura mater, and choroid plexus [34,35,36]. Similar to other organs, the population of brain border DC is heterogeneous under homeostatic conditions, comprised of pDC, cDC1, and cDC2, the latter of which are the most abundant subset [37].

A disease that results in CNS degeneration, MS occurs when T cells are activated against the protective myelin sheath that surrounds nerves, demonstrating the power of the immune system to target primary antigens from the CNS. Accordingly, much of the foundational knowledge about immune responses against CNS antigens originates from research carried out using a murine model of MS referred to as experimental allergic encephalitis (EAE). Not only do DC accumulate within the brain during disease progression, but they are also thought to be critical mediators of the disease pathology as a result of their antigen presentation to T cells via MHCII [37,38,39,40,41]. Despite controversy regarding the ability of microglia (MG) and BAM to present antigens to naïve T cells, elegant experiments involving conditional MHCII knockout on BAM, MG, and DC suggest that DC are the primary APC, unique in their sufficiency to drive EAE pathology [40,42]. While these studies do not exclude the possibility that MG and other cells with antigen presenting capacity are important in the context of brain tumors, they point to DC as essential mediators of CNS antigen presentation, especially considering the migratory capacity of DC compared to MG. In summary, the inflammation and immunogenic DC activity associated with MS sharply contrasts with the weak immune response against HGG.

## 4. Dendritic Cell Infiltration of HGG: Who Are the Players and Where Are They?

Precisely which factors govern the intensity of immune cell recruitment to the TME of HGG remains an outstanding question; however, the notion of immune involvement in HGG progression has been hypothesized since the 1970′s [43]. Two of the earliest observations of DC within rodent HGG models occurred over thirty years after the discovery of DC [44,45]. More recently, Carenza et al. utilized multi-parametric flow cytometry and single cell-RNA sequencing to examine the effects of corticosteroids, commonly administered to patients, on HGG-infiltrating DC. In addition to corticosteroid-dependent reduction of MHCII and CD40 on DC, they also identified a heterogeneous population of DC within patient HGG samples, including cDC1, cDC2, and moDC. cDC1 comprised the smallest portion, and significant levels of pDC were only detected in one out of a total of five untreated patients [46]. In a single cell RNA-sequencing study, Antunes et al. confirmed the reported DC heterogeneity in humans and extended these findings in a murine model of GBM. Interestingly, they uncovered further heterogeneity within the cDC2 population, with clusters of cDC2-expressing genes that correlate to cDC2a and cDC2b [47,48]. This diversity among the DC that infiltrate HGG has been verified by a similar study that utilized a de novo mouse model of HGG [49]. Moreover, Bowman-Kirigin et al. observed a comparable DC landscape in GL261 and CT2A murine brain tumors, with cDC1 and pDC present at much lower frequencies than cDC2 and moDC [50]. 

Although Bowman-Kirigin et al. determined that moDC comprise approximately one-third of GBM infiltrating DC, there is ongoing debate as to their source. In their study designed to profile the myeloid cells within HGG, Antunes et al. also investigated the contribution of circulating monocytes to myeloid populations, including DC. The proportion of proliferating DC within clusters correlating to cDC1 and cDC2 is the same in wildtype and CCR2 knockout mice, as measured by both single cell RNA-sequencing and BrdU incorporation [47]. While this is an indirect method to interrogate monocyte contribution, it suggests that intratumoral DC are generally not differentiated from monocytes. Fate-mapping studies would provide a more conclusive answer to the question of DC origin in this context. Interestingly, they identified a cluster of DC that express monocyte-related genes within human GBM samples [47]. On a similar note, pseudotime trajectory analyses performed by Friedrich et al. indicate a monocyte to DC pathway in patient samples of IDH mutant GBM that is dysfunctional in IDH wildtype GBM samples, adding further nuance to the heterogeneity of DC within HGG [51].

Although a plethora of studies speak to the presence of a variety of DC, these studies do not address their location within HGG and proximity to other immune cells. To examine spatial immune heterogeneity in patients, Kim et al. used flow cytometry and RNAseq analysis and found that the frequency of DC is higher in the periphery than in the core of HGG. Moreover, the expression of costimulatory markers and MHCII molecules by peripheral DC is decreased compared to expression by DC in the core of paired HGG samples, consistent with a reduced activation state of peripheral DC [52]. While replication of hypoxic conditions decreases expression of these markers in vitro, cytokines secreted by HGG and the immunosuppressive immune cells within the TME also likely contribute to this phenomenon in vivo [29,52].

Additionally, DC in the HGG periphery may have fewer interactions with T cells, which reside more frequently within HGG cores, further impeding the ability of DC to present Ag to local T cells. Moreover, the polarization of DC to be either immunogenic or tolerogenic depends on the signals received during antigen uptake, and if DC phagocytose tumor antigen within the hypoxic, immunosuppressive core, these signals are more likely to promote a tolerogenic rather than immunogenic response [52]. Although the differential localization of DC and T cells within HGG may reduce local T cell priming, DC located within the tumor periphery can still take up tumor antigen and migrate to the draining cervical lymph nodes through the meningeal lymphatics, a concept further explored below. One shortfall of using techniques such as flow cytometry and RNA sequencing on whole tumor excisions is that all information regarding the spatial organization of cells within the organ is lost due to homogenization. Since the leptomeninges is difficult to separate from the brain, any leptomeningeal DC present may contaminate what is being analyzed as the tumor (or the section of the brain bearing a tumor). Consequentially, more extensive characterization of DC localization using contemporary immune spatial profiling techniques will help inform hypotheses regarding their activation status and ability to interact directly with HGG and other infiltrating immune cells.

## 5. HGG Antigen Drainage for the Initiation of Peripheral Immune Response

As previously alluded to, DC-mediated antigen presentation and T cell priming in tumor draining LN are thought to be essential components of a successful antitumor immune response. However, the efficacy of HGG antigen drainage to cervical LN, specifically mediated by DC, remains a topic of study. The cervical lymph nodes are of particular interest due to their role as the primary draining lymph nodes of the CNS. Garzon-Muvdi et al. found that the OT-1 CD8^+^ T cells that infiltrate GL261-OVA brain tumors have already extensively proliferated. On the other hand, OT-1 cells in the draining cervical LN of the same mice exhibit CFSE labeling suggestive of clonal expansion [53]. Overall, this suggests that CD8^+^ T cell priming against tumor antigen primarily occurs in the cervical lymph nodes, and that after priming the activated T cells migrate back to the TME.

Two studies published in 2017 argue that T cell responses to intracranial tumors are limited because tumor antigen drainage through meningeal lymphatics is suboptimal. They demonstrated that lymphatic enhancement with VEGF-C elicits a more robust adaptive immune response against brain tumors. Song et al. utilized AAV-VEGFC-C and VEGF-C mRNA constructs to improve lymphatic vasculature in the brain and combined that with inhibitory checkpoint blockade (ICB). Although treatment with ICB alone modestly increase overall survival time and the frequency of tetramer-positive CD8^+^ T cells in cervical LN and tumors, the combination of VEGF-C and ICB resulted in a highly synergistic improvement in survival outcomes [54]. Concurrently, Hu et al. used a VEGF-C overexpressing GL261 model to further investigate the role of meningeal lymphatics in regulating brain tumor immunity with an emphasis on tumor antigen drainage via DC trafficking. In addition to yielding similar results regarding the combination of VEGF-C and ICB, they observed that the enhanced meningeal lymphatics result in increased DC migration from the CNS to cervical LN. This migration and associated improvement in survival depends on the CCL21/CCR7 chemotactic axis that DC use for migration to LN, indicating that DC-mediated antigen drainage is crucial for a robust antitumor response against HGG [55]. From the same group, Zhou et al. also showed that radiotherapy promotes a similar effect when combined with VEGF-C treatment in a GL261 glioblastoma model. This alludes to the idea that the immunomodulatory effects of radiation therapy that result in better CD8^+^ T cell HGG infiltration are strongly tied to tumor antigen drainage [56]. The necessity of enhanced DC-mediated antigen trafficking to LN in combination with either ICB or radiotherapy suggests that both DC and T cells are limited in function and depend on each other for their respective antitumor effector activities in an HGG setting. When formulating immunotherapeutic regimens, it will be useful to consider methods that promote immunogenic activation of both cell types.

Although the previous studies did not explore the subsets of DC that specifically phagocytose HGG antigens in the brain parenchyma, migrate to the cervical LN, and prime T cells, they both noted a resulting influx of CD8^+^ T cells to the tumors, suggesting a pivotal role for cDC1. Bowman-Kirigin et al. built upon this and the knowledge that cDC1 are essential for immune responses against other types of tumors to elucidate the role of cDC1 in priming an HGG specific CD8^+^ T cell response and mediating improved survival resulting from ICB. Despite the focus on cDC1 function, it was shown that all DC subsets phagocytose tumor antigens within GL261 and CT2A tumors that express the fluorescent molecule ZsGreen and that these DC are also present in the dura and cervical lymph nodes [50]. While these studies provide strong evidence for DC-mediated antigen trafficking to draining lymph nodes, it is interesting to note that there is also a small population of CD8a^+^CD103^+^ LN resident cDC1 that takes up fluorescent tumor antigens, leaving open the possibility that tumor antigen uptake by LN resident DC can contribute to the antitumor response [50]. That said, these findings corroborate the hypothesis that antigen presentation in draining LN is predominately mediated by DC from the CNS, rather than uptake of soluble CNS-derived antigens within the LN. 

## 6. Dendritic Cell Activity within HGG: Immunogenic or Tolerogenic?

Understanding the basis of DC suppression will guide the development of interventions that release DC from suppression to promote T cell priming. Pertaining to this, early in vitro experiments conducted by Kikuchi et al. suggest that HGG interfere with the maturation state of DC. Coculture of mature and immature cells with glioma cells does not change their phagocytic capacity but could diminish their surface expression of MHC and CD86. This decline in costimulatory molecules indicates a lack of DC maturation, separate from their ability to take up antigens. Coculture with glioma cells may also reduce IL-12 production by DC [57]. However, the notion that immature DC phagocytose antigens and do not secrete potent amounts of IL-12 is highly indicative of a tolerogenic character and sets a foundation for the possibility that DC may contribute to T cell anergy or dysfunction in an HGG setting.

A second set of coculture experiments performed by Wang et al. more specifically examines the relationship between HGG, Nrf2, and DC function. Nrf2 is a redox-sensitive transcription factor that initiates the transcription of genes involved in counteracting the harmful consequences of ROS, helping to maintain redox homeostasis [58,59]. Interestingly, Nrf2 expression in DC correlates with glioma progression in patients [60]. In vitro, glioma conditioned media upregulates Nrf2 expression in DC and diminishes their maturity as measured by CD80 and CD86 expression, repeating observations made by Kikuchi et al. These glioma conditioned DC showed additional signs of tolerogenic character as indicated by reduced IL-12 and increased IL-10 cytokines present in culture supernatant. Furthermore, DC pre-treatment with glioma conditioned media modestly diminishes T cell proliferation yet robustly reduces IFN-γ production by those same T cells [61]. In line with a tolerogenic DC profile, this indicates that the DC are not likely to prime naïve T cells in manner that induces a Th1 effector response. While siRNA knockdown of Nrf2 promotes glioma-mediated DC maturation and capacity to prime T cells, it also reduces phagocytic capacity in vitro, which was not observed in other studies [52,57,61]. This suggests that, while Nrf2 may play a crucial role inhibiting DC function in the context of HGG, other factors secreted by HGG likely affect DC activity independently of Nrf2. Though in vitro systems are conducive for answering specific questions in a closed system, use of an in vivo model to further probe the effect of HGG on DC would lend more support to these results. That said, preclinical murine studies utilizing ICB alone or VEGF-C meningeal lymphatic enhancement show an improvement in T cell immunity that mildly increases overall survival. Though this indicates that there are some DC capable of immunogenic antigen presentation, the lack of more robust tumor regression suggests that other factors prevent the initiation of a potent T cell response, one of which may be suboptimal DC activity.

## 7. Dendritic Cell Subsets and Their Role in ICB-Mediated Antitumor Immune Responses: cDC1 or cDC2?

In a more recent study, Simonds et al. compared the immune landscapes of ICB-responsive and refractory HGG from patient and mouse tumor samples. As expected, responsiveness to ICB positively correlates with HGG infiltration by cDC1, cDC2, and T cells, a factor that is highly dependent on the type of tumor or cell line. However, SB28 murine GBM tumors are ICB refractory in the brain but responsive when implanted subcutaneously [62]. Although the authors attribute this to poor antigen drainage from the brain to cervical LN, a more in-depth characterization of DC maturity, cytokine production, and costimulatory surface marker expression is needed to further elucidate the specific roles of DC subsets in priming T cells in draining LN and within the TME. Notably, an effective ICB-mediated immune response against subcutaneous SB28 tumors requires natural killer cells and CD4^+^ T cells but not CD8^+^ T cells, suggestive of a vital role for cDC2 instead of cDC1 [62]. In contrast, Bowman et al. used a cDC1-specific mouse model to more directly demonstrate that cDC1 are essential for an ICB-mediated immune response against GL261 and CT2A murine GBM tumors [50]. Although cDC1 cells are more closely associated with CD8^+^ T cell priming, they can also activate CD4^+^ cells [10]. This could reconcile the requirement for cDC1 observed by Bowman-Kirigin et al. with the necessity for CD4^+^ T cells demonstrated by Simonds et al. However, it is important to emphasize that these two studies utilized different cell lines to model GBM in mice, which may differ enough to stimulate two different immune responses upon treatment with ICB.

Given the variation in DC function between mouse models of HGG, it is essential to consider the differences between DC activity in these preclinical models and patient tumors. During brain tumor resections, tumor cells specifically take up and convert 5-aminolevulinic acid (5-ALA) into protoporphyrin IX (PPIX), a fluorescent molecule. Using PPIX as a surrogate for tumor antigen in patients with primary and recurrent GBM, Bowman-Kirigin et al. observed that tumor antigen is phagocytosed by both CD141^+^ and CD1^+^ cDC subsets, which correspond to murine cDC1 and cDC2, respectively. Interestingly, the CD141^+^ DC exhibit the highest frequency of tumor antigen uptake in patient samples, whereas moDC and cDC2 phagocytose the highest proportion of tumor antigen in mouse models [50]. Aside from the potential differences in DC subset function, popular murine models of GBM also tend to be more highly infiltrated by immune cells, which enhances their responsiveness to immunotherapies such as ICB. Ultimately, more studies are needed to examine the specific roles of various DC subsets, with special attention to how differences between mouse models and patients may affect the efficacy of immunotherapies.

## 8. Future Direction: How Can Basic Research Best Support Effective Translation?

The primary forms of immunotherapy currently being investigated and developed for treating HGG include ICB, the targeting of pro-tumor myeloid cells, vaccines against tumor antigens, and adoptive T cell therapy [63,64]. Despite promising preclinical results and hundreds of clinical trials, none of these immunotherapies have been approved for HGG treatment [65]. Aside from the aggressive and immunosuppressive nature of HGG, the small patient population and low level of patient participation limit the size of clinical trials, often leaving them underpowered [65,66,67].

Moreover, many patients receive standard of care treatments before or during enrollment in these clinical trials, yet the immunomodulatory effects of these treatments have not been conclusively determined. For example, radiation can have contradictory effects, reported to both heighten immunosuppression within the TME and initiate inflammatory antitumor responses in different models [68,69]. While TMZ temporarily curbs tumor progression and may serve as an adjuvant, it suppresses peripheral immunity and may inhibit the efficacy of therapies designed to boost peripheral antitumor immune responses [70]. Similarly, tumor resection drastically reduces tumor size and extends survival but may have a negative impact on immune responses, specifically regarding tumor antigen drainage through meningeal lymphatic vessels [71,72]. Interestingly, experimentally induced traumatic brain injuries (TBI) impair meningeal lymphatic function, and the same may be true of the damage caused to the meninges and brain during tumor resections [73]. If this is the case, tumor resections, and the ensuing inflammation, performed before immunotherapy administration may impede their efficacy. This is also important to consider since most preclinical mouse models of HGG involve drilling a small hole through the skull to allow for the stereotactic injection of tumor cells, a process similar to how experimental TBI are induced. A mouse model of genetically induced gliomagenesis provides an opportunity to bypass traditional implantation, warranting further investigation [74]. Because this common treatment for HGG may promote or inhibit a patient’s response to immunotherapies, it is essential to further study its effects to structure treatment regimens and clinical trials more successfully.

In addition to the standard treatments that new immunotherapies are often combined with as part of clinical trials, many of these new immunotherapies depend on DC activity. Successful in treating patients with non-small cell lung cancer and melanoma, ICB has proven mildly effective in preclinical GL261 preclinical mouse models of GBM [50,53,62]. However, GL261 brain tumors tend to be more highly infiltrated by T cells than human GBM, and thus, more likely to respond positively to ICB treatments than patients. DC dysfunction or suboptimal tumor antigen trafficking by DC may also contribute to the lack of ICB success in HGG patients. While T cell inhibition may be prevented, the T cells still need to be properly primed by costimulatory molecules and cytokines during antigen presentation by DC. Interestingly, administration of PD-1 blockade as a neoadjuvant before GBM resection extended median progression-free survival by twenty-seven days as compared to adjuvant PD-1 blockade after resection in a phase 1 clinical trial. Although this neoadjuvant ICB reinvigorated previously exhausted T cells and resulted in DC upregulation of interferon-stimulated genes, the authors attribute the failure to eliminate tumors to a result of the immunosuppressive myeloid cells that remained [75]. While these immunosuppressive myeloid cells may need to be targeted, an investigation of DC function beyond gene expression is required to more definitively evaluate DC as activated after neoadjuvant ICB [76]. Spatial analysis, especially with transcriptomics, is expected to provide granular information about the interaction between DC and other immune cell subsets in the TME. Such approaches are also expected to derive tremendous information from the rare and small amounts of patient HGG material that are commonly a limiting factor in analysis.

Vaccines against tumor antigens also rely on DC for efficacy. Although vaccines engage peripheral DC outside of the immunosuppressive HGG environment, there is some evidence to suggest that HGG can induce systemic changes of an immunosuppressive nature [77,78,79]. The extent to which these alterations may affect DC populations in the periphery is unknown. Likewise, tumor antigen-pulsed DC-based vaccines may also be affected by systemic and local immunosuppression caused by HGG or treatments such as chemotherapy. Despite the potential for DC suppression, a handful of clinical trials including DCVAX (NCT00045968), DENDR1 (NCT04801147), DENDR2 (NCT04002804) point to a potential for DC-based cancer vaccines [80,81]. Specifically, a phase III clinical trial recently demonstrated the efficacy of tumor antigen loaded DC vaccination as measured by improved overall survival in newly diagnosed and recurrent GBM patients. The immune response invoked by these antigen-loaded DC could be further improved by pre-conditioning of vaccination sites with tetanus toxoid to enhance DC migration to lymph nodes [81,82]. To bypass the invasive leukapheresis necessary to create individual DC-based vaccines, TLR agonists are an alternative treatment that could be used to activate and promote the maturity of DC that are already present within patients. Moreover, integration of either systemic or localized Flt3L to expand DC populations could be leveraged to enhance the efficacy of other DC-dependent immunotherapies [83]. As illustrated in Figure 2, many forms of immunotherapies considered for HGG treatment depend on DC activity, further arguing for more conclusive studies to determine the effects of HGG and standard treatments on local and systemic DC activity to better understand opportunities to improve current immunotherapeutic strategies.

## 9. Conclusions/Final Comments

DC are fundamental to antitumor immune responses as result of their ability to elicit a robust T cell response against tumor antigens. Although this paradigm is well-explored in the context of peripheral tumors, further investigation into the role of DC in mounting immune responses against intracranial tumors such as HGG is needed. Despite DC confinement to brain borders in the meninges and ventricles of healthy brains, a variety of DC populations infiltrate HGG. However, the extent of their infiltration and localization remains unknown and could affect the types of antigens they phagocytose (core vs. tumor periphery), the level of hypoxia they encounter, and their localization with T cells—all factors that could influence DC function and antigen presentation capacity. Outside of the local HGG environment, involvement of the peripheral immune system is critical for an effective antitumor immune response. The precise mechanics and location of CNS antigen capture is a topic of ongoing research; however, evidence from studies in the context of HGG points to local antigen uptake in the brain and DC-mediated trafficking as the primary means of eliciting T cell responses in draining LN. Additionally, the immunosuppressive factors secreted by tumor cells and pro-tumorigenic immune cells within the TME have been implicated in the polarization of DC to be tolerogenic rather than immunogenic. Although HGG TME are rich in highly immunosuppressive cytokines, studies into their effects on DC are largely limited to in vitro experiments. Interestingly, a handful of these in vitro experiments indicate that HGG suppress immunogenic DC activity, yet experiments involving improved lymphatic drainage for the use of ICB argue that DC function is sufficient to promote an adaptive response. However, the improved survival observed as a result of these experiments is often quite mild, possibly a consequence of substandard DC populations. It is critical to address these questions to better understand shortfalls of DC function in HGG and identify opportunities to integrate immunotherapies to optimize their efficacy.

## Figures and Tables

**Figure 1 cancers-15-02902-f001:**
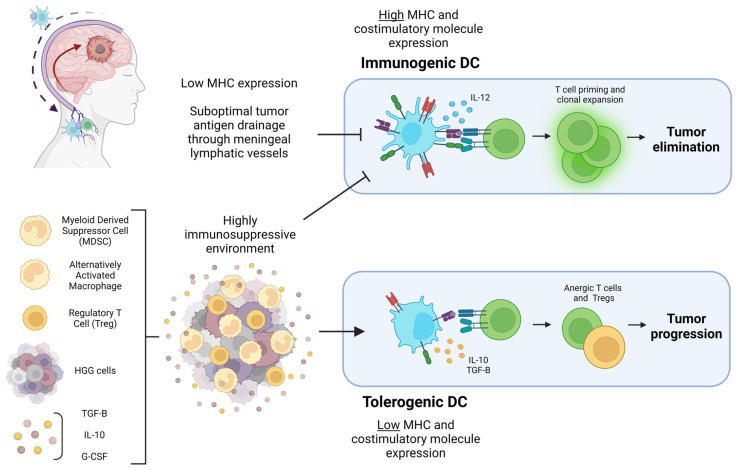
Diagram of the HGG-associated factors that can lead to tumor elimination or progression through DC polarization. Pictured in the top left corner is a diagram of HGG antigen drainage through meningeal lymphatic vessels to draining cervical lymph nodes mediated by DC. Suboptimal tumor antigen drainage prevents T cell priming and clonal expansion in the cervical lymph nodes. The highly immunosuppressive HGG TME pictured at the bottom left prevents DC from expressing the costimulatory molecules necessary for immunogenic T cell priming. Cytokines and immunosuppressive cells within the TME result in DC with a tolerogenic character that ultimately prime anergic T cells and Tregs, contributing to tumor progression. Figure created with Biorender.

**Figure 2 cancers-15-02902-f002:**
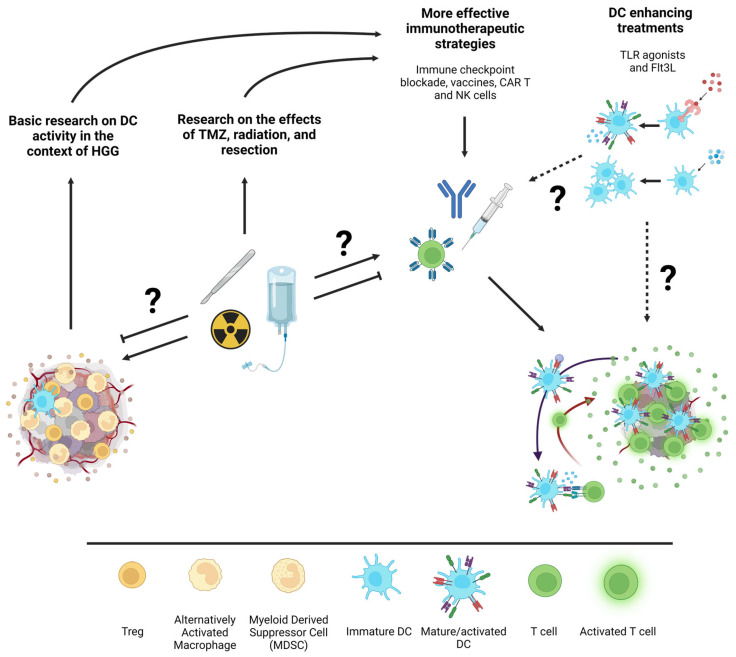
Schematic illustration of potential interactions between standard HGG treatments, immunotherapies, and DC enhancing treatments. From left to right, basic research on the infiltration and activation state of specific DC subsets within HGG and research on how standard of care treatments affect the immune response to HGG will inform more effective immunotherapeutic strategies. Additionally, the exploration of DC-specific forms of immunotherapies such as Flt3L to expand DC or TLR agonists to immunogenically activate them could further enhance immunotherapeutic strategies. Question marks are indicative of under-researched interactions between topics in the illustration. Figure created with Biorender.

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
