# Peer review of "Ready for Prime Time? Dendritic Cells in High-Grade Gliomas"

_cancers, 2023, doi:10.3390/cancers15112902_

Round 1
Reviewer 1 Report
I appreciated this review, well-written and well-organized. The topic is of great relevance and timely, especially from the translational point of view.
The is a place for small improvements listed as minor comments.
1. Recently advanced technologies based on single-cell and multi-omics allow us to rigorously characterize the tumor microenvironment by cataloging the different subsets of the immune context. The heterogeneity of infiltrating DC was previously described. For Instance, Pombo Antunes and colleagues published results obtained by performing a single-cell analysis (Nature Neuroscience 2022). The bioinformatic analysis revealed a heterogeneity of intratumoral DC, in both mice and humans.
The authors should consider and describe this important study and reconsider the more recent literature base on similar studies.
2. DCs are used as a medical product to treat patients affected by GBM. A number of data have demonstrated that DCs are safe and well tolerated. Results from a phase III study were recently published supporting the potential beneficial effects of this treatment (Liau et al JAMA Oncol 2023). Similar survival was observed in a much smaller clinical study named DENDR1 (NCT04801147). Additionally not only newly diagnosed GBM but also recurrent GBM patients were treated with DCs (DENDR2 and V-DENDR2, NOA 2019), considering the positive effects of TT as preconditioning, as previously described by Mitchell and colleagues (Nature 2015).
The authors should consider the contribution of DC commenting on these studies but also look for other studies in which DCs were administrated to patients.
Author Response
We appreciate you taking the time to carefully evaluate our manuscript and are grateful for the insightful feedback. Your recommendations and suggested references are very timely as to current research in the field, and we feel like our manuscript is much improved by incorporating these. Thank you.
- Recently advanced technologies based on single-cell and multi-omics allow us to rigorously characterize the tumor microenvironment by cataloging the different subsets of the immune context. The heterogeneity of infiltrating DC was previously described. For Instance, Pombo Antunes and colleagues published results obtained by performing a single-cell analysis (Nature Neuroscience 2022). The bioinformatic analysis revealed a heterogeneity of intratumoral DC, in both mice and humans.
The authors should consider and describe this important study and reconsider the more recent literature base on similar studies.
Although this paper was addressed in the original version of the Antunes et al manuscript regarding the potential origin of DC (specifically their CCR2 KO experiments), we agree that it is important to include information about the DC heterogeneity that this group observed using single-cell analysis. We have added this to Section 4 (Lines 177-180), near the reference to a similar study performed by Carenza et al. Additionally, we searched through more recent literature and decided to reference a study by Yeo et al that used similar methods to explore DC heterogeneity in a different model of GBM.
- DCs are used as a medical product to treat patients affected by GBM. A number of data have demonstrated that DCs are safe and well tolerated. Results from a phase III study were recently published supporting the potential beneficial effects of this treatment (Liau et al JAMA Oncol 2023). Similar survival was observed in a much smaller clinical study named DENDR1 (NCT04801147). Additionally not only newly diagnosed GBM but also recurrent GBM patients were treated with DCs (DENDR2 and V-DENDR2, NOA 2019), considering the positive effects of TT as preconditioning, as previously described by Mitchell and colleagues (Nature 2015).
The authors should consider the contribution of DC commenting on these studies but also look for other studies in which DCs were administrated to patients.
Yes, although our review is more focused on intracranial DC that infiltrate HGG, it is important for our review to address the topic of DC as a medical product. Thank you for bringing these studies and the point about preconditioning with TT to our attention—we have referenced each of them in Section 8 (lines 411-421). Because peripheral DC as a medical product our separate from the central focus of this review on intracranial DC in the context of HGG, we have decided to limit references to similar studies. However, it raises a thought-provoking question about which aspects of immune suppression may be preventing more complete tumor control by the immune system after successful treatment with antigen-pulsed DC (and how those immunosuppressive mechanisms can be countered in combination with DC administration).
Reviewer 2 Report
The subject of study is extremely interesting for the development of different strategies for a possible effective treatment for gliomas and, as stated by the authors themselves, little is known about the interaction of immune system cells with the cells of this tumor. A better understanding of this interaction and the molecular mechanism that hinders a robust immune response is crucial for the advancement of immunotherapies. Despite thinking that the review could go deeper into the molecular mechanisms of the interaction of immune cells with glioma cells, I believe that the review is interesting, with a more general view of the problem, both for researchers in the field of cancer immunology and for researchers who study gliomas and for those who intend to take an interdisciplinary approach. therefore, I indicate the approval of the article with some small considerations: authors need to make a minor grammatical revision, with special attention to the parts that describe some published experiments. The texts of lines 129-131 and lines 222-229 are difficult to understand the findings of the cited articles.
The writing in general is good and easy to understand, however, as written earlier, some parts need to be better explained so that the understanding of the findings of the cited experiments is better understood
Author Response
We appreciate you taking the time to carefully evaluate our manuscript and are grateful for the insightful feedback. Your recommendation to take a closer look at some of the language used to describe the findings of cited articles was helpful, and we feel like our manuscript is much improved by doing so. Thank you.
The subject of study is extremely interesting for the development of different strategies for a possible effective treatment for gliomas and, as stated by the authors themselves, little is known about the interaction of immune system cells with the cells of this tumor. A better understanding of this interaction and the molecular mechanism that hinders a robust immune response is crucial for the advancement of immunotherapies. Despite thinking that the review could go deeper into the molecular mechanisms of the interaction of immune cells with glioma cells, I believe that the review is interesting, with a more general view of the problem, both for researchers in the field of cancer immunology and for researchers who study gliomas and for those who intend to take an interdisciplinary approach. therefore, I indicate the approval of the article with some small considerations: authors need to make a minor grammatical revision, with special attention to the parts that describe some published experiments. The texts of lines 129-131 and lines 222-229 are difficult to understand the findings of the cited articles.
Yes, although we included a handful of references to coculture experiments in which the effects of glioma cells on DC were examined (lines 284-308), we agree that it would be interesting to further delve into the molecular mechanisms of the interactions between immune and glioma cells. Interestingly, there is not a lot of data on this topic in the literature, and most of it was produced over a decade ago. As such, we think it may be an appealing direction for further research with modern techniques.
Upon going back to the lines that you pointed out, we can see how they are challenging to understand and may interrupt the flow of the paper. We have revised lines 129-131 and 222-229 to more clearly communicate the point that we were attempting to relate. Please find these changes below:
Before: “Although HGG grew when exclusively implanted in the brains of experimental rodents, they were rejected when implanted in the periphery or co-transplanted in the brain and a peripheral location”
After: “In these early experiments, tumors grew when implanted into the brains of immunocompetent rodents but were rejected when implanted in the periphery, inferring location dependent immunogenicity. Interestingly, when co-transplanted in the brain and a peripheral location, both tumors were rejected, indicating that extracranial immune activity can participate in intracranial tumor rejection.”
Before: “Garzon-Muvdi et al found that adoptively transferred OT-1 CD8+ cells within implanted GL261-OVA brain tumors are terminally divided. On the other hand, OT-1 cells in the cervical LNs exhibit variation in the cycles of division according to CSFE labeling, suggestive of CD8+ T cell clonal expansion. However, this is not exclusive to the cervical LN as there are also small proportions of less divided cells present in tumors [54]. Thus, there remains the possibility that tumor-specific CD8+ T cell priming primarily occurs within draining cervical LN while an additional, likely secondary, level of priming may occur within the TME.”
After: “Garzon-Muvdi et al found that the OT-1 CD8+ T cells that infiltrate GL261-OVA brain tumors have already extensively proliferated. On the other hand, OT-1 cells in the draining cervical LN of the same mice exhibit CFSE labeling suggestive of clonal expansion. Overall, this suggests that CD8+ T cell priming against tumor antigen primarily occurs in the cervical lymph nodes, and that after priming the activated T cells migrate back to the TME.”
Reviewer 3 Report
Ref: Cancers- ID: cancers-2355041
Journal: Cancers-MDPI
The manuscript entitled: “Ready for prime time? Dendritic cells in high-grade gliomas.” by Conarroe & Bullock is an interesting review article summarizing the presence and function of dendritic cells within the tumor microenvironment of high-grade gliomas with consideration of therapeutic opportunities. This is a well written review article that fits within the scope of the journal and the special issue. Some revisions are needed for being able this manuscript to be accepted for publication in the Cancers Journal.
Please find below the comments-suggestions that will help the authors improve the current version of this manuscript:
Major/minor comments:
Introduction: -It would be also good to include the new WHO2021 Classification of brain tumours within the context of HGG here.
-Line 104-105: “Overall, DC represent a heterogenous population of cells with both im-munogenic and tolerizing potential within the TME..”: please expand further on this DC heterogeneity and subpopulations and its importance for improvement current treatments.
-Lines 120-121: you have mentioned IL-10 twice; please amend with the correct cytokine.
-Section 3: “Multiple Sclerosis/Experimental Allergic Encephalomyelitis: what are DC capable of in the CNS?”: it is not very clear why this part needs to be a separate since does not directly reflect the main goal of the manuscript(HGG). I would suggest to incorporate this part in another one than having this as a separate section. Also, why the authors have chosen multiple sclerosis and not other neurodegenerative diseases?
-lines 198-199: please define which are those soluble factors secreted by HGG and the immunosuppressive immune cells within the TME that are also likely contribute to this phenomenon and describe a bit the in vivo mechanism here.
-Lines 227-228: “tumor-specific CD8+ T cell priming primarily occurs within draining cervical LN while an additional, likely secondary, level of priming may occur within the TME.”: please explain more this notion and the reason why you focus on cervical LN and not other types here.
-Figure 1 and 2 should be cited within the text in terms of the information being given.
-Figure 1: please expand on the figure legend by explaining briefly the mechanisms identified in the figure. Also please define the term of alternatively activated macrophages-are they TAMs, M1 or M2?
-Lines 342-343: please explain more why we would expect to see differences between mouse models and patient DCs and how this may affect the efficacy of DC-based immunotherapies related to these studies.
-In the last part of future directions, the authors could discuss more the impact on new technologies such as single-cell and spatial omics/multiplexed imaging providing information that these technologies may be involved to improve immunotherapies and the future personalized medicine for HGG.
-Figure 2: could have been more enriched in terms of the types of immune cells/molecules and TME that is affected under the treatments. Also, a more explanatory figure legend would further help the readers understand the meaning of the interaction of the different treatments and the main message that needs to be given here. This figure needs to be revised.
-Figure legends should include the term at the end:” Figure created with Biorender”.
Author Response
We appreciate you taking the time to carefully evaluate our manuscript and are grateful for the detailed and insightful feedback. Your recommendations are thoughtful and timely as to current research in the field, and we feel like our manuscript is much improved by incorporating these. Thank you.
Introduction: -It would be also good to include the new WHO2021 Classification of brain tumours within the context of HGG here.
Yes, this was noted and added to the introduction (lines 24-25).
-Line 104-105: “Overall, DC represent a heterogenous population of cells with both im-munogenic and tolerizing potential within the TME..”: please expand further on this DC heterogeneity and subpopulations and its importance for improvement current treatments.
While we expand upon the heterogeneity and importance of specific subpopulations later in the paper (Sections 4, 5, and 6), we can see how what was meant to be a conclusion actually makes the reader more curious for details on this. Thus, we have added the phase in italics below to clarify that these topics will be touched upon in more detail.
“Overall, DC represent a heterogeneous population of cells with both immunogenic and tolerizing potential within the TME and make logical immunotherapeutic targets due to their ability to promote antitumor immunity, as described in more detail below.”
-Lines 120-121: you have mentioned IL-10 twice; please amend with the correct cytokine.
This has been corrected, thank you for catching this detail.
-Section 3: “Multiple Sclerosis/Experimental Allergic Encephalomyelitis: what are DC capable of in the CNS?”: it is not very clear why this part needs to be a separate since does not directly reflect the main goal of the manuscript(HGG). I would suggest to incorporate this part in another one than having this as a separate section. Also, why the authors have chosen multiple sclerosis and not other neurodegenerative diseases?
Yes, upon further reflection we agree that there does not need to be a separate section about MS. We have revised this section to emphasize our original point about DC having the potential to generate a robust immune response against CNS antigen (as exemplified by MS). In doing so we changed the title of the section and reduced the references to MS-specific studies to half of the original, with the aim of justifying the emphasis on DC as the primary APC in the CNS (as compared to microglia or macrophages).
-lines 198-199: please define which are those soluble factors secreted by HGG and the immunosuppressive immune cells within the TME that are also likely contribute to this phenomenon and describe a bit the in vivo mechanism here.
This line was a reference to the immunosuppressive cytokines originally mentioned in lines 120-121. We see how this was vague, and as a result we changed the wording to clarify this. Because the in vivo mechanism was already described in the manuscript, we will refrain from repeating that again here.
-Lines 227-228: “tumor-specific CD8+ T cell priming primarily occurs within draining cervical LN while an additional, likely secondary, level of priming may occur within the TME.”: please explain more this notion and the reason why you focus on cervical LN and not other types here.
We have decided to remove this sentence due to the confusion it leads to and because it also distracts from the importance of antigen drainage to and T cell priming within the cervical lymph nodes. Although we address the importance of cervical lymph nodes as the draining lymph nodes for HGG, we appreciate that this may not have been clear enough. To address this, we added the following phrase:
“The cervical lymph nodes are of particular interest due to their role as the primary draining lymph nodes of the CNS.” (Lines 232-233)
-Figure 1 and 2 should be cited within the text in terms of the information being given.
Yes, we have made these corresponding changes.
-Figure 1: please expand on the figure legend by explaining briefly the mechanisms identified in the figure. Also please define the term of alternatively activated macrophages-are they TAMs, M1 or M2?
The figure legend has been updated with explanations of the mechanisms pictured. The alternatively activated macrophages are TAMs, and as such they are pictures within the TME in Figure 1. To make this clear I have added a description to the figure legend.
-Lines 342-343: please explain more why we would expect to see differences between mouse models and patient DCs and how this may affect the efficacy of DC-based immunotherapies related to these studies.
Although I can see that this was not clear, lines 356-358 reference a difference in DC subset activity between a murine model and patient samples. To clarify, I have added the following sentence:
“Aside from the potential differences in DC subset function, popular murine models of GBM also tend to be more highly infiltrated by immune cells, which enhances their responsiveness to immunotherapies such as ICB.” (Line 358-360)
I have also chosen to revise my concluding statement about the efficacy of DC-based therapies to be less exclusive. I had meant that differences in the general immune response (and potentially in DC responses) could affect the efficacy of a broad array of immunotherapies.
-In the last part of future directions, the authors could discuss more the impact on new technologies such as single-cell and spatial omics/multiplexed imaging providing information that these technologies may be involved to improve immunotherapies and the future personalized medicine for HGG.
We find this to be an interesting point and future direction of study, particularly spatial omics. Accordingly, we opted to include the following sentence in Section 8 (lines 412-416):
“Spatial analysis, especially with transcriptomics, is expected to provide granular information about the interaction between DC and other immune cell subsets in the TME. Such approaches are also expected to derive tremendous information from the rare and small amounts of patient HGG material that are commonly a limiting factor for analysis.”
-Figure 2: could have been more enriched in terms of the types of immune cells/molecules and TME that is affected under the treatments. Also, a more explanatory figure legend would further help the readers understand the meaning of the interaction of the different treatments and the main message that needs to be given here. This figure needs to be revised.
This figure has been revised to include a better explanation of the immune cell types in the illustration. We have also added a more extensive explanation of the interactions between research areas, treatments, and outcomes.
-Figure legends should include the term at the end:” Figure created with Biorender”.
Thank you for letting us know this. It has been added to both figure legends.
Round 2
Reviewer 3 Report
Ref: Cancers- ID: cancers-2355041
Journal: Cancers-MDPI
The manuscript entitled: “Ready for prime time? Dendritic cells in high-grade gliomas.” by Conarroe & Bullock is an interesting review article summarizing the presence and function of dendritic cells within the tumor microenvironment of high-grade gliomas with consideration of therapeutic opportunities. This is a well written review article and has now been improved after the revisions. The authors addressed all the revisions adequately and I recommend the manuscript to be accepted for publication in the Cancers Journal.